# Potassium Control of Plant Functions: Ecological and Agricultural Implications

**DOI:** 10.3390/plants10020419

**Published:** 2021-02-23

**Authors:** Jordi Sardans, Josep Peñuelas

**Affiliations:** 1CSIC, Global Ecology Unit CREAF-CSIC-UAB, 08913 Bellaterra, Catalonia, Spain; josep.penuelas@uab.cat; 2CREAF, 08913 Cerdanyola del Vallès, Catalonia, Spain

**Keywords:** potassium battery, plant physiological control, water transport, nutrient and metabolite transport, limitation, stress response

## Abstract

Potassium, mostly as a cation (K^+^), together with calcium (Ca^2+^) are the most abundant inorganic chemicals in plant cellular media, but they are rarely discussed. K^+^ is not a component of molecular or macromolecular plant structures, thus it is more difficult to link it to concrete metabolic pathways than nitrogen or phosphorus. Over the last two decades, many studies have reported on the role of K^+^ in several physiological functions, including controlling cellular growth and wood formation, xylem–phloem water content and movement, nutrient and metabolite transport, and stress responses. In this paper, we present an overview of contemporary findings associating K^+^ with various plant functions, emphasizing plant-mediated responses to environmental abiotic and biotic shifts and stresses by controlling transmembrane potentials and water, nutrient, and metabolite transport. These essential roles of K^+^ account for its high concentrations in the most active plant organs, such as leaves, and are consistent with the increasing number of ecological and agricultural studies that report K^+^ as a key element in the function and structure of terrestrial ecosystems, crop production, and global food security. We synthesized these roles from an integrated perspective, considering the metabolic and physiological functions of individual plants and their complex roles in terrestrial ecosystem functions and food security within the current context of ongoing global change. Thus, we provide a bridge between studies of K^+^ at the plant and ecological levels to ultimately claim that K^+^ should be considered at least at a level similar to N and P in terrestrial ecological studies.

## 1. Introduction

In recent years, the role of potassium in terrestrial ecosystems has been increasingly studied, thus observing that its role in determining plant growth, species composition, and ecosystem function can be of similar importance to the role of the much more studied elements nitrogen (N) and phosphorus (P) [1,2,3,4,5]. While N and P are mostly present in forming part of biomolecules, K is present in living organisms mostly as a free cation (K^+^). K^+^, together with calcium (Ca^2+^), are the two most abundant inorganic chemicals in plant water cellular media, and K^+^ is the second most abundant nutrient in leaf biomass after N, which highlights its great involvement and unavoidable contribution to plant functioning. At the plant-community level, K^+^ also limits community growth [4,5]. Recent reports have observed great direct impact of potassium in plant photosynthetic capacity [6] and growth [7] in complex plant functional mechanisms in responses to different stresses [8,9] and plant homeostasis [10] and metabolic control [11]. Since our past review of the role of K^+^ in terrestrial ecosystem responses to global change drivers [4], a number of reports have allowed us to establish a better link between potassium and responses to global change drivers [12,13,14] but also with terrestrial ecosystem functions, and structural variables such as growth and nutrient cycling [7,15,16]. Here, we synthesize these important roles, from the metabolic and physiological functions in individual plants to the complex roles in terrestrial ecosystem functions and food security while considering ongoing global changes. We thus provide a bridge between the studies of K^+^ in plants and the studies of K^+^ in ecosystems, and ultimately claim that its introduction in terrestrial ecological studies should be at least at a level similar to those of N and P.

## 2. Stable Cation in Solution Necessary for Plant Functional Homeostasis and Production Control

### 2.1. The Biogeochemical Properties of K^+^

Granitic rocks are the most abundant rocks in continental plates, and feldspars are principal constituents of granite. Hydrolysis of the most abundant feldspars releases K^+^, Na^+^, and Ca^2+^. Further, these cations are present in considerable concentrations in organisms. K^+^ is the most abundant cation in plants (non-halophytes) because it is required for a variety of plant functions [17,18,19,20,21]. K^+^ is an activator of >60 plant enzymes (e.g., in glycolysis and the Krebs cycle) [22,23]. Na^+^ can substitute K^+^ in some roles, such as osmotic adjustment in some organisms, but excessive Na^+^ concentrations in cells can be detrimental [18]. Ca^2+^ is also an abundant and important cation in key cellular processes, but for several cases in which Ca^2+^ is involved in responses to external stimuli, the responses are dependent on the movement of K^+^ across membranes [24,25,26]. Unlike K^+^, Ca^2+^ is also involved in the formation, composition, and stabilization of cell walls [27]. Both K^+^ and Ca^2+^ are the cations involved in more plant functions and are indispensable in plant nutrition, but K^+^ limits plant growth more frequently than Ca^2+^ [28,29].

K^+^ has the largest ionic radius of the three main cations released from feldspar hydrolysis (Ca^2+^, Na^+^, and K^+^); thus, its electrical density and electrostatic interactions, which decrease with the charge–radius ratio of the ion [30], are the lowest of the three cations. This implies that K^+^ has less interference capacity with biological functions in cytoplasm cell media, as the strength of the solvation bond with water molecules (coordination complex, chelate formation) is lower, and its mobility capacity in water solution is higher than that of Na^+^ and Ca^2+^ [31,32]. Thus, K^+^ is a more stable cation in cell water solutions than Na^+^ and Ca^2+^ and is the most appropriate cation to control vital functions, such as hydraulic transport, transmembrane transport, cellular osmosis maintenance, or movement linked to changes in turgor, without unwanted interference. An increasing number of recent reports have provided evidence for K^+^-mediated control in most plant functions [4,33,34]. Evidence elucidating the role of K^+^ in basic ecological processes, such as competition and other biotic relationships, has also been increasingly reported [4,35,36].

### 2.2. The Multiple Functions of K^+^ in Plants

The role of K^+^ in photosynthetic activity and plant growth is fundamental and complex for multiple direct and indirect mechanisms [6,37,38]. Adequate K^+^ supply enhances photosynthetic assimilation, improves nutrient uptake [7], and maintains adequate leaf inclination by turgor control [39]. K^+^ has a fundamental role in stomata-opening control, allowing adequate gas and water fluxes [6,38]. Adequate K^+^ concentration in chloroplasts is also necessary to facilitate a well-structured stroma lamella in chloroplasts, thereby supporting chloroplast integrity and light-absorption efficiency [7,40]. In this context, K^+^ deficiency has been observed to strongly suppress rubisco biosynthesis and activity [41,42]. These effects are partially beyond the role of K^+^ in ribosomal transport and movements, which influence the rates of protein synthesis [43]. Thus, K^+^ controls rubisco activity by affecting its biosynthesis; chloroplast stability; and nutrient, water, and gas fluxes [6].

K^+^ is present in plants in concentrations ranging from 50 to 150 mM in the cell cytoplasm and vacuole [44]. The concentration of K^+^ in the cytoplasm is typically constant (approximately 50 mM), while that in the vacuole may vary substantially [44]. In addition to directly affecting the control of osmotic pressure, K^+^ concentration is involved in indirect processes of plant-cell homeostasis control. In plants, epidermal cells generally rely heavily on inorganic ions (mainly K^+^) for osmotic adjustment [45]. Conversely, in mesophyll cells, the total contribution of organic osmolytes is much higher [46], although K^+^ remains the dominant osmoticum [46,47]. Mesophyll K^+^ homeostasis is maintained at the expense of the K^+^ supply coming from the epidermis cells, where cytosolic K^+^ may decline to very low levels [48]. K^+^ is a major contributor to osmotic adjustment at the beginning of the water deficit, but its role is overtaken by organic osmolytes with growth transitions [49]. Plants also have a large capacity to redistribute absorbed K^+^ between vacuolar and cytosolic pools to ensure cytosolic homeostasis [50].

In summary, K^+^ has a key role in the mechanism that controls the transport of water, metabolites, and nutrients across plant tissues and organs; plant defense against oxidative stresses; and maintenance of osmotic homeostasis [10,11,51].

## 3. Plant K Uptake Mechanisms

### 3.1. The Role of Transporters and Channels

Plant K^+^ uptake is determined by sophisticated mechanisms of genetic expression in response to stimuli, such as drought, and by the redistribution of absorbed K^+^ between vacuolar and cytosolic pools to ensure cytosolic homeostasis [52]. Plants have developed several strategies for managing K^+^ when soil-available K^+^ is scarce or fluctuating. All terrestrial plants have a complex mechanism for K^+^ uptake from soil and cellular transport [53]. Several families of membrane protein transporters are involved in K^+^ uptake, allocation, and homeostasis [8]. K^+^ transporters belong to several families, including KT/HAK/KUP, Trk/Ktr/HKT, and CPA. Some members of these transport families mediate the transport of K^+^, some are involved in the transport of Na^+^, and some contribute to the transport and uptake of both Na^+^ and K^+^. The cation preferences of these transporters depends on affinity and structural properties. These transporters can act as low-affinity transporters (LATs) that operate at high soil external (e.g., soil) K^+^ concentrations or as high-affinity transporters (HATs) that operate at low external K^+^ concentrations [54]. Further, membrane LAT proteins, such as AKT1 and various nonselective channels, have been identified; the latter operates at very high external K^+^ concentrations [55,56]. High-affinity K^+^ transporters are active proteins that are saturable and act at low K^+^ concentrations (<1 mM) [4]. High-affinity transporters (HATs) operate at low soil K^+^ concentrations. Identified HATs are KT/HAK/KUP transporters, such as HAK1 in rice, pepper [57], and barley [58]; HAK5 in *Arabidopsis* [59], barley [58], and tomato [60]; and KUP1 and KUP2 in barley [58]. The mechanisms of nonselective channels and active transporters of K^+^ in roots can be distinct among different plant species [22], and also can exist in a different number of genes encoding for the same family of transporters among distinct plant species [61]. Furthermore some studies have observed that individual transporters may have both high- and low-affinity activity, for instance, AtKUP1 in *Arabidopsis thaliana* [62]. Moreover, in addition to the KT/HAK/KUP high-affinity transporters, other transporter families are also involved in K^+^ transport in plants. HKT transporters act as Na^+^-K^+^ symporters in plants and have an outstanding role in salt tolerance in plants due to maintenance of K^+^ uptake by K^+^-Na^+^ cotransport under saline stress [63]. Moreover there is solid evidence that some electroneutral cation–proton antiporters, the CAP1 and CAP2 families, are also involved in nonchannel K^+^ transport [64]. The NHX1 involved in K^+^ vacuole loading is an example of the CAP1 protein family [64]. More recently, eight isoforms of NHX proteins have been described that are located in the cell membrane and in diverse organelle membranes, mainly vacuoles and endoplasm, in *Arabidopsis,* where they allow the interchange of K^+^ from cytosol to organelle with H^+^ from organelles to cytosol [65]. CHX proteins from the CPA2 family are involved in homeostasis pH regulation through the counter-exchange of K^+^ and H^+^ [64,66]. Several studies have shown that CHX proteins comprise a Na^+^/H^+^ exchange domain, but are also involved in K^+^ transport [65] being involved in salt stress response by maintaining pH K^+^ homeostasis, an effect observed in different plant organs such as roots and leaves, and also in pollen [67,68,69,70]. The K^+^ efflux antiporter (KEA) proteins are another CPA2 family. For instance, in *Arabidopsis* there are six distinct KEA transporters with a higher preference for K^+^ to cation/H^+^ exchange [71]. This CAP2 family has been described in the inner envelope membrane of plastids, thylakoids, and Golgi organelle, and has been linked to pH and K^+^ homeostasis in endomembrane organelle compartments and is upregulated by K^+^ deficiency, salinity, and osmotic stress [72,73].

However, more than 50% of overall plant K^+^ uptake contribution is related to K^+^ membrane proteins acting as channels [74]. In this case, the transported compound K^+^ crosses across channel when some physicochemical variable reaches a certain level of concentration, changing the channel conformation, and thus allowing the passage of K^+^ [75]. Besides the reported K^+^ transport capability of some nonselective cation channels (NSCC), it is believed that plants have evolved three different types of K^+^ channels: voltage-gated shaker-type channels (AKT, KAT, SKOR, GORK), voltage-independent two-pore K^+^ channels (TPK) and the K^+^ inward-rectifier (Kir) channels. As commented in this review, voltage-gated shaker channels are involved in K^+^ transport at several levels, from K^+^ uptake from soil to stomata openness control; however, there are also two-pore K+ (TPK) channels, independent of voltage but selective to cytosolic Ca^2+^ and regulated by calcium-dependent protein kinases, that selectively mediate the release of K^+^ from vacuoles [76]. Finally, the inward-rectifier K^+^ channels (K_ir_, IRK) have a pore domain similar to that of voltage-gated channels and are activated by phosphatidylinositol 4,5-bisphosphate (PIP_2_). These channels allow an inward movement of K^+^ under hyperpolarization, but do not allow the exit of K^+^ under depolarization [77].

### 3.2. Plant Morphology and Ecological Aspects in Plant K^+^ Uptake

These mechanisms to control plant K^+^ uptake are very important ecologically. Besides the above-mentioned role and control of transporters and channels, root growth and architecture are also strongly linked to K^+^ uptake in plants. K^+^ influences root growth and architecture by several mechanisms, such as root cell expansion and root hair growth [7]. More intensive root-system proliferation was found in K^+^-efficient genotypes in various plant species, including important crops such as rice [40], maize [78], and *Medicago truncatula* [79]. Some species are also capable of facilitating K^+^ uptake under K^+^-limiting conditions by activation of high-affinity K^+^ transporters [60,80,81]. From these data, it becomes clear that enhanced root-system proliferation must be supported by more effective K^+^ uptake to some extent. In *Arabidopsis thaliana* and tobacco, the length and number of first-order lateral roots decrease, while the number of second-order laterals increases in K^+^-depleted plants [82]. Different species can also have distinct capacities of proliferation of lateral roots into K^+^-rich soil patches [7], mainly mediated by AKT1 high-affinity K^+^ transporters [83], with all of them relating root architecture with soil K^+^ availability and plant K^+^ status. All these differences in K^+^ uptake among plant species can be crucial in the ecosystem’s responses to global change drivers. These drivers favor those community species with the most adequate K^+^ uptake capacity to respond to changing environmental circumstances.

High-affinity AKT channels and HAK transporters are the main systems of plant K^+^ uptake under low K^+^ soil availability [76]. Observed membrane hyperpolarization was the first evidence of a root response to low soil K^+^ concentrations [84] (Figure 1). This hyperpolarization is coupled with H^+^ extrusion and extracellular acidification, which then triggers cellular mechanisms to take up K^+^, with the direct effect of membrane hyperpolarization driving the influx of K^+^ by transporters [85]. Low levels of soil K^+^ can induce an increase in cytoplasmic reactive oxygen species (ROS) [86] and Ca^2+^ concentrations by ROS-activated Ca^2+^ channels mediated by hyperpolarization [87,88].

Thus, ROS and intracellular Ca^2+^ signaling interact with each other as ROS production is increased by NADH-oxidase production by cytosolic Ca^2+^ concentration; thus, ROS activates Ca^2+^ influx channels [89,90]. However, these increases in cytosolic Ca^2+^ trigger signaling pathways that increase the synthesis of high-affinity K^+^ transporters such as HAK [91,92]. During the first hours of stress, a series of equilibrium changes involving K^+^ tend to facilitate plant recovery [52]. Current experimental studies have observed that high-affinity K^+^ transporters are mediators of K^+^ stress responses [93]. Thus, membrane hyperpolarization increases when soil K^+^ availability is low [21]. Simultaneously, there is an upregulation and activation of high-affinity selective K^+^ transporters by triggering Ca^2+^ signaling mechanisms [93,94,95], as the membrane potential between root cells and soils is the main source of energy that allows the uptake of K^+^ against K^+^ concentration gradients [26] (Figure 2). Moreover, membrane hyperpolarization also activates voltage-dependent K^+^ channels, such as the shaker AKT1-like channels, which in some plant species are considered the main membrane proteins that mediate K^+^ influx into roots channels [26]. Some channels, including AKT1 in *Arabidopsis* [96], OsAKT1 in rice [97], ZMK1 in maize [98], SKT1 in potato [99], or TaAKT1 in wheat [100], have voltage sensors that change the 3D configuration and are then activated under hyperpolarization by changes in membrane potential [96,97,98,99,100]. Importantly, NH_4_^+^ competes with K^+^ by high-affinity voltage-dependent transporters, but not by high-affinity voltage-dependent channels in plant uptake from soil, such as those observed in several species [76].

## 4. Internal Transport of K^+^, Water, Nutrients, and Biomolecules

### 4.1. The Control of Internal Water Transport

One key role of K^+^ as a crucial nutrient in terrestrial ecosystems is as the main contributor to water movement and solute transport [4,101] by controlling transmembrane potentials and osmotic pressure. Moreover, K^+^ root concentrations affect osmotic pressure in the xylem (root pressure), which drives and favors the long-distance flow of sap from roots to shoots [101].

The relationship between K^+^ and aquaporins is another key mechanism for controlling hydric status, water movement, osmotic potentials, and intercell transport. Aquaporins reside in the plasma membrane and the tonoplast, and play important roles in plant–water relations by facilitating the transport of water across biological membranes and regulating osmotic potential and hydraulic conductivity [102]. The large number of aquaporin types has been explained by their importance in regulating plant metabolic processes under various physiological states and environmental conditions [103]. Aquaporins regulate the internal redistribution of mineral nutrients by transporting them from the endoplasmic reticulum to the plasma membrane via the Golgi apparatus, as well as undergoing repeated cycles of endocytosis and recycling through the early endosome to the multivesicular body and/or prevacuolar compartments, before eventually being targeted to the vacuole [104]. As K^+^ is the major osmolyte, its uptake and movement throughout plant tissues and organs is accompanied by water flux through the aquaporins. Thus, there is a strong positive link between K^+^ absorption and water uptake [103]. A mutual interconnection at the transcription level between aquaporins and K^+^ transporters has been observed [105], and the transcripts encoding aquaporins were strongly affected by K^+^ starvation, even without water stress [106]. Aquaporins can function as turgor sensors to modulate the conductance of K^+^ channels [103]. In line with studies of *Arabidopsis*, we identified 12 aquaporin genes in the shoots and 15 in the roots that were significantly upregulated after K^+^ resupply [107]. In onion roots, water transport is sensitive to inhibitors of aquaporins and K^+^ channels, and a decrease in hydraulic conductivity after K^+^ channel inhibitor treatment indicates that K^+^ fluxes are involved in aquaporin activity in the plasma membrane [108]. Moreover, as observed in *Arabidopsis* roots, the same channel inhibitor (CsCl) inhibited the expression of genes encoding water channels and aquaporins and inhibited high affinity K^+^ transporter 5 (HAK5) [108]. In rice, the expression of the K^+^ channel activator phosphatidylinositol 4,5-bisphosphate responded similarly to K^+^ deficiency and water stress, suggesting that aquaporins and K^+^ channels are functionally coregulated during cell turgor regulation [108] During drought stress, plants modulate their water and ion-uptake capacities by regulating aquaporins and K^+^ channels at the transcriptional level to respond to water deficiency [109,110]. K^+^ has important passive (osmotic effect) and active roles in the transport of water and xylem sap from roots to leaves [111]. These processes have been associated with K^+^ accumulation in the pectic matrix of intervessel pits of the xylem, suggesting that K^+^ actively promotes pit opening, thus facilitating the flux of xylem sap [111].

### 4.2. The K^+^ Role in the Complex Mechanisms of Internal Biomolecules, Nutrients, and Energy in Plants: “The Potassium Battery”

The role of K^+^ in plant transport is not limited to the control of the movement of water from roots to shoots by simple indirect osmotic force; K^+^ also directly controls phloemic transport [112] (Figure 2). An increase in photosynthetic activity in leaves increases the transcription of genes for the formation and assembly of membrane channels for K^+^ transport and the photosynthate-induced phloem K^(+)^ channels (AKT2 K^+^) that regulate transmembrane potentials and transport. K^+^, together with high saccharide concentrations resulting from higher photosynthetic activity, facilitates the transfer of saccharides from parenchymal cells to the phloem [113]. The concept of the “potassium battery” [32] refers to a transmembrane K^+^ gradient used by plants as an energy source that is transported via the phloem stream, storing and providing energy to other transport processes (Figure 2). This has been associated with the general K^+^ transport capacity among various plant structures [113].

Plant tissues with sufficient energy and ATP availability can load K^+^ to the phloem, which reaches a notable concentration, and the sap flow transports K^+^ to organs requiring it; K^+^ can also return to roots [114,115]. This transport creates a K^+^ gradient between the apoplast and sieve phloem-accompanying cells, as well as an energetic mobile gradient along the phloem, which acts as a useful energy store for alleviating local energy limitations; this phenomenon facilitates the activation of protein transmembrane transporters [34]. This energy can be used by AKT2 K^+^ channels, which can shift an ATP-independent mechanism that helps remote tissue cells reload carbohydrates or release K^+^ and nutrients to phloem. The created transmembrane K^+^ gradient can thus be conducted by phloem flux and thereafter from the cytosol of the sieve phloem-accompanying cells to the apoplasts and cytosols of foliar, shoot, and root cells, and it can be used to transport metabolites through the openings of specific protein transmembrane channels, such as AKT2-like channels, by the passage of K^+^ across the membrane [34] (Figure 2). In this way, plant cells of different tissues can exert a remote control of transport processes of nutrients, including K^+^ and carbohydrates, by locally regulating membrane ATPases and the AKT2 K^+^ transporter function [116]. The control of the ATK2 K^+^ transporter mode of function is crucial in this complex mechanism. The sensitivity and capacity of this mechanism seems to be distinct among plant species, and can even be stimulated differently by different biotic and abiotic circumstances and variables [117], as another K^+^-related plant function that can have a great ecological importance, with ample room for further research.

Depending on the local energy status and gradients of H^+^, K^+^, and other nutrients, the AKT2 K^+^ transporter can work in mode 1, pumping K^+^ from the apoplast to the phloem; or in mode 2, allowing the passage of K^+^ between these two compartments [34,116]. K^+^ is loaded into the phloem by ATP consumption in source tissues, and it then constitutes an energy gradient (potassium battery) that assists membrane ATPases-H^+^ transporters in the transmembrane gradients required for transportation between phloem and all tissue cells. However, the control mechanism of the shift of the AKT2 K^+^ transporter from mode 1 to 2 and vice versa remains unclear and merits future research to reach a full understanding of the role of K^+^ in plant functions. Recently, Cuin et al. [117] proposed the existence of authentic action potentials in plants similar to those of animals based on K^+^ membrane transport systems. The outward-rectifying potassium-selective channel (GORK) limits the amplitude and duration of action potentials, while the weakly rectifying channel AKT2 K^+^ affects membrane excitability.

Thus, K^+^ is a critical resource involved in the control of water and nutrient fluxes from roots to different plant organ tissues and organic molecules among distinct plant organ tissues, controlling cell osmosis, turgor, and pH, and allowing adequate cell organelle status and movement. In terrestrial biological systems, potassium has reached a pivotal role in the coordination and control of the relationships and fluxes between the internal plant and external plant environments. Thus, potassium plays a key role in the ecophysiological and ecological changes of plants in responses of plant communities to environmental shifts, such as climate change.

## 5. Role of K in Plant Responses to External Environmental Conditions

Movements of K^+^ across the membranes of cells or organelles are also responsible for the final response in the cascade of effects from stimulus to response in plants. Adequate cellular K^+^ concentrations are necessary for plants to respond properly to various types of stress, such as drought, salinity, flooding, or herbivores [26,118,119].

### 5.1. Abiotic Stress

The transcriptional and post-transcriptional control of K^+^ passive and active membrane transporters is a central topic of current research, owing to their important role in plant development, functionality, and control of responses to environmental stresses [20,120,121] (Figure 1). Internal chemical signals of stress, such as high concentrations of ROS or Na^+^ and/or low concentrations of H_2_O or O_2_ due to metabolic failure under several stresses (drought, flooding, or salinity) and the activation of membrane “osmosensors” (e.g., high Na^+^ concentrations in saline soil) associated with shifts in membrane potential [121], are responsible for the upregulation and activation of voltage-dependent membrane K^+^-efflux transporters. These transporters transfer K^+^ from roots to soil and can also directly inhibit low-affinity transporters (LATs) [122,123]. This process during the early stages of stress has a negative effect on the capacity of plants to cope with stress, as plants with suboptimal K^+^ concentrations have a lower transcriptomic control of the genes involved in antistress mechanisms [118].

However, several other parallel stress mechanisms allow K^+^ replenishment, and thus enhance stress tolerance [26,124]. Further, external or internal stimuli (e.g., drought, changes in cellular osmotic pressure or pH, changes in metabolite concentrations, or excess light or oxidation) mediate responses by activating genes encoding hormones and/or proteins that can catalyze the transcription of genes that produce high-affinity K^+^ channels (HAT) [125,126]. The resultant increase in cellular K^+^ concentration consequently stimulates the expression of antistress genes or directly stimulates the production of antistress proteins from their intermediates [121]. For example, some experimental studies have reported that high K^+^ concentrations enhance the transcriptional response to anoxic stress [121] or salinity and drought stress [120]. However, these mechanisms are not sufficient for returning cellular K^+^ concentrations back to normal under extreme and persistent stressful conditions, especially under saline conditions, which are necessary to inhibit the mechanisms of transcription of genes encoding proteases and endonucleases [127], which can lead to cellular and plant death [128]. Mechanisms for maintaining adequate K^+^ in root cells require the downregulation of the outwardly rectifying K^+^ channel (SKOR) involved in transport from roots to xylems [129], ensuring sufficient root K^+^ concentrations to allow root growth [130].

An adequate supply of potassium can increase the production of some amino acids—primarily those serving as precursors for important protective molecules, including proline under metabolic and physiological stress conditions [11,131,132]. Proline accumulation has been related to rapid recovery after stress, as it can provide energy during the recovery processes [133]. Practically, there are counterbalance mechanisms between K^+^ and organic osmolyte concentrations in plant cells. The reduction of free sugars as osmolytes under high cellular K^+^ concentrations is related to the synthesis of large biomolecules. Thus, the number of small molecules, such as free sugars, amino acids, organic acids, and amides, are reduced in the cell, while the concentration of phenols increases; these compounds aid in plant resistance to stress [134] and facilitate plant responses to abiotic stress [85]. Carbohydrates, mostly in terms of hexose content, are decreased in leaves due to a sufficient K^+^ supply and transported to another plant organ owing to better phloem activity. In contrast, K^+^ deficiency results in decreased activity of pyruvate kinase and/or increased invertase activity, which reduces the concentration of starch in leaves, owing to starch synthase inhibition [135]. Each plant genotype can adapt to drought stress by producing higher numbers and concentrations of amino and organic acids in the cell solution to account for the lack of K^+^ for osmotic adjustment, coupled with decreased plant growth and reserve accumulation under K^+^ deficit [136].

However, the role of K^+^ in maintaining adequate plant-cell osmosis pressure and pH homeostasis has been demonstrated to be very complex and at work even at the cell organelle level. For instance, under hyperosmotic conditions, several plant species accumulate free fatty acids in mitochondria, which activates PmitoK(ATP), a mitochondrial potassium channel of K^+^, from the cytoplasm to mitochondria, thus allowing sufficient turgor in the organelle to maintain homeostasis [137]. Moreover, cooperation, equilibrium, and control feedback mechanisms between PmitoKATP and the K^+^/H^+^ antiporter allow K^+^ fluxes to function as pH homeostatic mechanisms [138]. Under salt stress, higher K^+^ levels diminish lipid peroxidation, and the synthesis of antioxidant enzymes increases the growth of tomato plants, thus indicating the involvement of K^+^ in antioxidative stress mechanisms [139]. Higher cellular concentrations of K^+^ are associated with the oxidative stress recovery and with a better performance of the antioxidant enzymes, such as ascorbate peroxidase, dehydroascorbate reductase, glutathione reductase, superoxide dismutase, catalase, peroxidase, and NADPH oxidase [140,141]. Potassium deficiency is most likely the primary reason for the increased NADPH oxidase and NADPH-dependent ROS formation, owing to the generation of ABA. This capacity of K^+^ to counteract ROS formation has been found to be particularly intense in response to ROS formation from salt stress. The external use of potassium in a saline growing medium was shown to improve salt tolerance by mitigating ROS formation in *Triticum aestivum* [142,143], *Zea mays* [144], and *Oryza sativa* [145].

Under drought, K^+^ transporters and channels have several regulatory mechanisms; for example, there are several environmental variables, including availability of soil water and nutrients, involved in the control of the activity of K^+^ transporters in vacuole membranes (TPK channels) or the membranes of stomata guard cells (HAT and LAT transporters, and GORK channels) [146]. Sensing and signaling during osmotic stress due to water deficits are pivotal for plant–water status and lead to rapid changes in gene expression; this stimulates the synthesis of abscisic acid (ABA) and inhibits the synthesis of auxin [147,148]. As observed in *Arabidopsis*, in guard cells, the tonoplast-localized K^+^/H^+^ exchangers NHX1 and NHX2 are pivotal in the vacuolar accumulation of K^+^, and nhx1 and nhx2 mutant lines are dysfunctional in stomatal regulation [38]. K^+^ accumulation is a requirement for stomata opening and a critical component in K^+^ homeostasis, which is essential for stomata closure, suggesting that vacuole K^+^ fluxes are also crucial for regulating the vacuole dynamics and luminal pH that underlie stomata movements [38]. The intracellular events that underlie stomata opening begin with plasma membrane hyperpolarization caused by the activation of H^+^-ATPases, which induces K^+^ uptake through voltage-gated inwardly rectifying K^+^ in channels [149]. In plants, H^+^-ATPases belong to the multigene family of the P-type ATPases, with 11 genes in *Arabidopsis*, which are all expressed in guard cells [150]. In guard cells, the action of H^+^-ATPase activity is positively regulated by blue light and auxins, whereas Ca^2^^+^ and ABA act as negative regulators. Hyperpolarization leads to K^+^ uptake via the activation of inward K^+^ rectifying channels (KAT), thereby increasing their osmotic potential and driving the uptake of water that generates the turgor pressure necessary for cellular expansion and growth [151,152]. Potassium uptake is accompanied by the electrophoretic entry of the counter-ions chloride, nitrate, malate, and sulphate, as well as by the synthesis of malate, depending on the environmental conditions and the time of day [153,154]. These osmolytes, together with sucrose accumulation, increase turgor in guard cells, and thus incite stomata opening. Existing K^+^ accumulation to drive rapid stomata opening at dawn is essential for the sucrose-dominated phase. This indicates that sucrose replaces K^+^ for turgor maintenance in the afternoon, rather than simply enhancing stomata opening.

ABA increases the function of high-affinity selective K^+^ transporters in the membranes of root cells and of inward GORK transporters in foliar stomata, and decreases the synthesis of auxin [148]. Lower auxin concentrations favor stomata closure. Thus, water losses are avoided, and resistance to several stresses such as drought or salinity is favored. The up- and downregulation of the K^+^ transporter system dependent on ABA and auxin are thus very important in the change from stressful-to-normal and from normal-to-stressful conditions [148]. Stomata closure is initiated by the activation of localized chloride and nitrate efflux channels in the plasma membrane (SLAC1 and SLAH3) that are regulated by the SnRK2 protein kinase OST1 and the Ca^2^^+^-dependent protein kinases CPK21 and 23 [155,156]. CPK6 also activates SLAC1 and coordinately inhibits rectifying K^+^ in channels to hinder stomata opening [157,158]. Sulphate and organic acids exit the guard cell through R-type anion channels. The accompanying reduction in guard-cell turgor results in stomata closure [149].

ABA is synthesized in plastids and cytosol—mainly in the vascular parenchyma cells, but also in guard cells—through the cleavage of a C40 carotenoid precursor followed by a two-step conversion of the intermediate xanthoxin into ABA via ABA-aldehyde [159,160]. The increases in ABA in stomata guard cells under stressful conditions mediate the equilibrium of stomata opening and closure (Figure 1). The mechanism of ABA-induced stomata closure has been the subject of numerous studies. Further, it is known to be driven by a decrease in guard-cell turgor due to K^+^ efflux and associated anions, such as Cl^−^ and/or malate, which are triggered by an increase in cytoplasmic Ca^2+^ concentrations [161]. ABA triggers Ca^2+^ transporter activity and increases Ca^2+^ concentrations in stomata guard cells. Thus, Ca^2+^ concentrations act as signaling mechanisms that increase membrane depolarization and activate GORK K^+^ transporters (guard-cell outward-rectifying K^+^ channel) [162] and Cl^−^ transporters; thus, the concentrations of K^+^ and Cl^−^ in the cytosol decrease [50]. This membrane depolarization also promotes TPK K^+^ channel activity in vacuole membranes, with a consequent transfer of K^+^ from the vacuole to the cytosol to outside the guard cells, promoting the turgor loss and stomata closure [50]. This inhibition of H^+^-ATPase was linked to anion-channel activation, resulting in membrane depolarization. Anion channels such as rapid channels (R-type) and slow channels (S-type) facilitate the efflux of malate^2^^−^, Cl^−^, and NO_3_^−^ [163]. The decreased level of malate in guard cells is also linked with the gluconeogenic conversion of malate into starch [164]. Jasmonates interact with the ABA pathway by increasing the influx of Ca^2^^+^, which stimulates the cascade and closes the stomata [165].

Plant responses under more favorable conditions, such as an increase in resource-abundant and high-light and/or elevated auxin concentration conditions, can stimulate the transcription of genes encoding specific K^+^ channels [113]. KAT channel activation favors increases in cellular K^+^ concentrations, and thus stomata opening [166]. Recently, BAG genes, which mediate and facilitate the synthesis of KAT1 at the plasma membrane, have been identified in *Arabidopsis thaliana* [12]. This offers new possibilities for improving plant resistance to drought and pathogens. Auxins typically play a positive role in stomata opening, but high auxin concentrations can inhibit stomata opening [167]. Low auxin concentrations activate inward K^+^ channels, leading to stomata opening, whereas high auxin levels promote outward K^+^ channels while inhibiting inward K^+^ KAT channels, resulting in stomata closure [167]. Auxins and cytokinins inhibit ABA-induced stomata closure by enhancing ethylene production, as observed in some plant species such as *Arabidopsis* [168]. Incident light radiation, mainly high-energy blue radiation, can also open stomata by activating inward K^+^ KAT channels [169].

Furthermore, the KAT1-like inward K1 channels in guard cells can be also modulated by molecules other than hormones (ABA and auxins). Some polyamines produced during some stresses also target and modulate stomata movements, thus linking stress conditions, polyamine levels, and stomata regulation. Polyamines mimic stress conditions in blocking stomata opening and inducing stomata closure. Plants under stress conditions accumulate higher levels of polyamines, including putrescine, spermidine, spermine, and cadaverine [170], which can also directly inhibit KAT1-like inward K1 channels in guard cells and thus stomata opening [170].

K^+^ also plays a key role in the response to oxidative stress. The link between stimulus and final response associated with K^+^ transport has also been observed in antioxidant metabolism under stress and the maintenance of plant growth under these conditions [52,112,120,148]. Plants that are exposed to environmental stresses show enhanced K^+^ requirements; furthermore, they exhibit increased oxidative damage to cells via ROS formation, particularly during photosynthesis [21]. Under drought conditions, excess ROS production may increase cellular lipid peroxidation, leading to an increase in cellular membrane permeability, as evidenced by increases in the electrolyte leakage and malondialdehyde (MDA) content [171,172]. Soleimanzadeh et al. [173] performed an experiment with sunflower (*Helianthus annuus* L.) and reported that an adequate supply of K^+^ significantly decreased MDA content under water-shortage conditions, which clearly indicated the role of K^+^ in mitigating oxidative stress.

Potassium upregulates antioxidant metabolism and alleviates growth inhibition under water and osmotic stress. K^+^ is associated with the accumulation of osmolytes and an increase in antioxidant components in plants exposed to water and salt stress [13,119]. In this case, sufficient K^+^ supply is significant, as K^+^ is a key factor in promoting the tolerance of plants to various stresses by the cumulative effect of several attributes under the direct control of gene expression. K^+^ promotes various pathways and mechanisms, including increasing the accumulation of organic osmolytes, enhancing enzyme activities, and maintaining a higher K^+^/Na^+^ ratio [15,174,175,176]. Thus, maintaining adequate K^+^ levels in plant tissues indirectly helps mitigate ROS formation through adequate regulation of stomata movements, osmoregulation, and water-use management [54]. K^+^ also reduces ROS generation by inhibiting NADPH oxidase [174], leading to reduced lipid peroxidation [177]; the result is the peroxidation of membrane lipids by ROS under salinity [178] and water stress [179].

The supply of K^+^ results in an increase in the growth and activity of antioxidant enzymes in both normal and stressed plants [179]. Under both normal and stressed conditions, K-fed plants experience significant increases in the synthesis of osmolytes, such as free proline, amino acids, and sugars, indicating significance in growth under water-stress conditions. Wheat plants accumulating greater K^+^ were able to counteract the water-stress-induced changes by maintaining a lower Na^+^/K^+^ ratio [179]. Owing to the impairment of photosynthetic CO_2_ fixation, plant molecular O_2_ under stress conditions increases ROS production within the plant cell [21,180], resulting in the degradation of the photosynthetic pigment and cellular membranes. It has been experimentally observed that adequate K^+^ availability under drought stress reduced photosynthesis inhibition by mitigating ROS formation [181,182]. Thus, it was suggested that an adequate supply of K^+^ under drought conditions improved photosynthetic CO_2_ fixation and the export of photosynthates from source to sink organs [183,184] and prevented photosynthetic electron transport to O_2_. Hence, ROS formation was reduced [21,180]. The direct link between potassium starvation and oxidative stress has also been observed in tomato plants, showing a rise in ROS species and strong alteration in anti-ROS enzyme concentrations under K^+^ starvation [177].

The transport mechanisms involved at the same time in K^+^ and Na^+^ transport play a key role in determining adaptation to salt stress by determining Na^+^/K^+^ ratio in different plant compartments [185,186,187]. Different studies suggest that salinity increase K^+^ circulation in vascular tissues; for instance, by the upregulation of AKT2/3 channels in phloem and SKOR channels located in stellar root tissue [188,189]. K^+^ causes a shift of energy from biosynthetic processes to defensive, and uptake repairmen tissue under salinity [186]. However, as commented, in early stages of salinity, the overexpression of root GORK channels can increase K^+^ losses and trigger programmed cell death [190].

The metabolic and physiological mechanisms linked to cellular K^+^ concentrations improve the response to drought and salt stress and enhance growth under these stressful conditions, as demonstrated by the addition of K^+^ in several crop experiments [119]. Further studies of K^+^ availability and stressful conditions (e.g., drought) and metagenomic and transcriptomic analyses are warranted to achieve a more complete and general understanding of the control of K^+^ movement and the functioning of K^+^ transporters in intra- and intercellular membranes.

### 5.2. Biotic Stress

The roles of K^+^ in plant responses to pests and diseases and the underlying mechanisms are more complex. In particular, K^+^ deficiency in plants has been associated with lower cell-membrane resistance and higher concentrations of sugars and amino acids, which can increase risks of pathogenic and herbivorous damage [37]. K^+^ deficiency has also been associated with higher synthesis and concentrations of jasmonic acid (JA), and therefore with the enhancement of defensive mechanisms and the activity of high-affinity selective K^+^ transporters in the cells of fine roots [35] (Figure 1). Direct responses to various stimuli, such as light or pressure (e.g., from animals), mediate the movement of leaves and flowers [191]. Protein-membrane receptors (such as cytokinin receptor Crel) [192] can change their tertiary structures and activate a cascade of phosphorylation by messenger proteins that ultimately activate K^+^ and Cl^−^ channels; this allows an increase in turgor in some cells and a decrease in others, directing the movement of leaves or flowers [191]. The rapidly expanding “omic” resources and molecular techniques are expected to facilitate the identification of proteins responsible for the steps involved in K^+^ movement in plants [112].

The mechanisms of these processes imply a complete array of signaling cascades, with multiple feedbacks and steps of gene transcription that are not well known. Thus, K^+^ is a central regulator of plant function, mediated in most cases in the sequence: stimulus -> receptor -> response (Figure 1 and Figure 2).

## 6. Role of K in Terrestrial Ecosystems

### 6.1. Ecological Aspects of the K^+^ Cycle in the Plant–Soil System

The upscaling role of potassium in terrestrial ecosystem function, from its atomic–ionic properties and soil traits to its control of water and nutrient fluxes in global soil–plant systems, has been scarcely investigated. K^+^ is present in three distinct components of soil: dissolved in soil water, adsorbed onto particles of clay and organic matter, and held within the crystal structures of clay particles or primary silicates, such as feldspar and mica. The organic matter in soils contains a negligible amount of K^+^, as K^+^ is not a constituent of biomolecules and is easily and rapidly leached from organic matter because of its high solubility [4]. A small soil potassium proportion (0.1–0.2%) of the total soil potassium is in solution and immediately available for plants, whereas the K^+^ exchangeable fraction is 1–2%, and the unavailable fractions are 91–99% [193,194]. The available form is easily leached by runoff; hence, the amount of K^+^ that is accessible to plants is often lower than that of N or P, despite the amount of K^+^ occasionally being higher than that of N and especially of P [195,196].

Despite the great solubility of K^+^ [31,32], terrestrial ecosystems have a large retention capacity, consistent with a strong evolutionary importance of the capacity to retain K^+^ to ensure its availability. Most (96–99%) of the K^+^ in soils is in the crystalline structures of micas and feldspars, and is thus not available to plants [194]. The K^+^ in soil solution, which is directly available to plants, comprises a minor proportion (0.1–0.2%) of total soil K^+^, whereas the exchangeable (labile) K^+^ adsorbed on clay and organic matter constitutes 1–2% of the total soil K [194]. The proportion of available versus total soil K^+^ is thus lower than that of N and P [36]. The K^+^ availability depends on any type of clay content (mainly esmectites) in soil and the level of leaching. In most cases, inceptisols and entisols are particularly rich in clays and not excessively leached. Alfisols are rich in clay content and also are not excessively leached, and thus can have a good K^+^ availability, whereas mollisols, which have a high humus concentration with very large nutrient retention capacity, are the soils most likely to have the highest K^+^ availability [197]. Plants can actively retain K^+^, increasing its availability through various mechanisms, such as hydraulic lift [198,199]. Plants can use hydraulic lift to transport K^+^ released from mineral weathering from deeper soil layers to shallow layers [198,199]. Thus, the capacity of soil minerals to release K^+^, hydraulic lift, and the capacity of K^+^ resorption allow most ecosystems worldwide to increase the capacity of K^+^ retention and the availability of K^+^ for microbes and plants [198,199].

### 6.2. K^+^’s Role in Plant–Soil System Shifts under Distinct Abiotic and Biotic Environmental Circumstances

There is scarce information on the intra- and interspecies variability in the capacity of plants to respond to direct or indirect (e.g., low soil K^+^ mobility in dry soils) K^+^ soil-availability scarcity and the role of root transporters in the capacity of plants to adapt to this scarcity. Moreover, some differences among species in high-affinity transporters and the number of genes encoding them [22,61] and in the capacity to vary root growth and architecture to uptake K^+^ [7,83] have been observed, facts that merit future research to obtain more in-depth information on plant communities’ shifts under environmental changes. Plants that have genes encoding for K^+^ high-affinity transporters HAK1 and HAK5 synthesis resist and adapt better to drought conditions than plants without these genes [16,200], with the consequent increase in root growth mediated by an increase in auxin synthesis [16]. Other K^+^ transporters, such as KUP7, have also proved to improve root K^+^ uptake efficiency and drought-stress adaptation [201]. This demonstrates the pivotal role of K^+^ plant uptake capacity, even in low K^+^ availability conditions, to better adapt to water-stress conditions. This is not a trivial question, given the projected increase in dry periods in several areas of the world [202] that could also affect crop species. The role of plant K^+^ uptake and use efficiency linked to root transporters should be considered in ecology studies of the resistance of plant species and communities to the projected increase in drought in many regions of the world. However, the role of K in mitigating drought impacts on plant functions at the individual level under more arid future scenarios in semiarid regions, such as the Sahel or Mediterranean areas, and at the ecosystem level, remains to be analysed. 

The role of K^+^ in several ecological and ecophysiological processes can be also influenced by N eutrophication, increasing atmospheric CO_2_, or changes in land cover, but K^+^ itself can play a significant role in the response/adaptation of terrestrial ecosystems to these global change drivers. This possible key role remains to be studied at local, regional, and global scales [4]. The sophisticated mechanisms in which K^+^ is involved are pivotal for water- and nutrient-use efficiency, stress avoidance, and general plant homeostasis control, altogether implying several ecological consequences for the ecosystem, such as species replacement or altered water and nutrient cycling. For example, in the drought-tolerant tree species *Olea europea,* when soil temperature reaches 37 °C, its root growth is inhibited, as is K^+^ uptake, with a drop of stem K^+^ concentration, water content, and use efficiency [203]. The sophisticated mechanisms in which K^+^ is involved are pivotal for water- and nutrient-use efficiency, stress avoidance, and general plant homeostasis controls that imply several ecosystem-ecological consequences, such as species replacement or water- and nutrient-cycle variations in global change scenarios. As observed in several forests, such as temperate and tropical forests, K^+^ is rapidly released during litter decomposition [204,205,206]. However, more than 70% of the initial K^+^ content can be released during the first week of the decomposition process [207]. In fact, K^+^ is the most rapidly released nutrient from litter during decomposition, even faster than other highly soluble nutrients, such as Mg^2+^ and Ca^2+^ [207,208]. This is consistent with the frequently observed K^+^ limitation for soil microbes that can easily absorb K^+^ from the decomposing soil organic matter [209]. In this context, there are several bacterial clades with high capacity to resorb K^+^ from soil aggregates and clays and solubilize it, thus increasing K^+^ availability [210,211]. Further, plants can improve their capacity for K^+^ uptake by mycorrhization [212,213,214] and root exudates [215,216]. Some studies have suggested that plants can invest C in root exudates [215,216] or in root mycorrhization [217,218,219], specifically to cope with K^+^ soil limitation. However, other studies have not observed a mycorrhization rise related to plant K^+^ limitation.

A crucial ecological aspect of K^+^ is how plants can adapt to low-K-availability soils. Current experimental studies have observed that high-affinity K^+^ transporters are mediators of K^+^ stress responses [220]. Higher plants can adapt to drought stress by producing higher numbers and concentrations of amino acids and organic acids in the cell solution to replace the lack of K^+^ for osmotic adjustment coupled with its negative effect on plant growth and reserve accumulation [213].

The capacity of K^+^ to stimulate plant growth is directly related to its role in maintaining cellular turgor [148,221] and indirectly to its role in controlling the osmotic potential of stomata guard cells and thus stomata opening [166]; this is also related to interactions between cellular K^+^ concentrations and their feedbacks and links with the synthesis of ABA and auxin [148] (Figure 1). Moreover, Grefen et al. [222] observed that the secretion of material for wall remodelling and cell expansion is dependent on the interaction of the soluble *N*-ethylmeleimide-sensitive attached protein. This phenomenon mediates vesicle fusion with the cell membrane simultaneously and binds with the voltage-dependent-sensors of cell-membrane K^+^ channels to achieve the transmembrane voltage necessary for secretory traffic in parallel with K^+^ cell uptake. Finally, the capacity of K^+^ to stimulate plant growth rates is due to its role in maintaining cellular turgor, its control of the osmotic potential of stomata guard cells, and thus stomata opening, and its interactions with the synthesis of ABA and auxin [148,166,221] (Figure 1). Several studies have suggested that the activity of key enzymes related to plant production capacity, such as nitrate reductase, RuBisCO, starch synthase, sucrose phosphate synthase, amylase, invertase, phosphofructokinase, and pyruvate kinase, significantly depends on the K^+^ sufficiency of plants [223,224,225].

Despite this, few studies in field conditions have focused on the importance of K^+^ in ecosystem production or in limitations of plant growth potential, particularly compared with studies focused on the importance of N and P limitation and their roles in ecosystem production and in their potentials in structure and function. The existing metadata analyses suggest that the percentage of terrestrial ecosystems that are K-limited or K-colimited with N and/or P is high (around 69%), and have detected at least some significant limiting role of K^+^ [4,226,227]. Potassium limitation can affect photosynthesis production despite no N or P limitations, mostly through its role in stomata control and water fluxes [228,229]. Similarly, the increase in N and P supply in forest growth is considerably improved with further addition of K^+^ [230].

Most macroecological studies identifying patterns of foliar nutrient concentrations as a function of global environmental traits have focused on N and P and their ratios [231,232]. Far fewer studies have focused on K [4]. Some studies on the relationship between terrestrial plant stoichiometry and climatic gradients observed that foliar K^+^ concentrations are negatively correlated with mean annual precipitation (MAP) [232], whereas others observed contrary relationships [233,234]. This apparent contradiction can be explained by the great differences in plant uptake capacity and use efficiency depending on the climate conditions where plants are naturally adapted. Leaves of species adapted to arid sites are small, with higher K^+^ concentrations than those of wetter sites [235]. Furthermore, tree species adapted to dry sites, such as Mediterranean evergreen and dry tropical forests, have a higher capacity to change their seasonal internal allocation of K^+^, with a higher allocation of K^+^ to leaves during summer (the driest season) than the species at wetter sites [234,236,237].

Climate models project an increase in the extent of drought over large areas of the world, such as the Mediterranean Basin and the Sahel [238]. Drought can increase total soil K^+^ but decrease its soluble soil fraction by increasing aridity; warming hinders potassium uptake and transport and, consequently, plant growth; this indicates the significant impact of drought conditions on plant performance [239]. Globally, plant K^+^ uptake appears to be associated with water availability [233,234,240]; lower K^+^ uptake negatively affects water uptake by reducing the activity of aquaporins [183]. These interconnected relationships strongly suggest a cascade of a higher water deficit, lower K^+^ uptake, and a reduced capacity to avoid drought, which could cause significant problems in dry regions that are threatened by future increasing dryness and drought. These scenarios may be even worse in some regions if torrential rains increase and MAP decreases, as projected for areas such as the Mediterranean Basin and the Sahel [202].

Thus, current data suggest that K^+^ uptake is strongly determined by soil water availability [241,242] and vice versa, and that K^+^ is crucial to the strategies of water uptake and economy by increasing water use efficiency and limiting water loss, thus under more aridity, a negative feedback between water and K^+^ is expected [4]. K^+^ uptake is favored in wet ecosystems, as well as in dry ecosystems where plants have adapted to drought with acquired mechanisms to counterbalance water scarcity by increasing K use efficiency through conservation mechanisms, such as increasing K^+^ resorption [243]. Thus, K^+^ losses under drought observed in field conditions are a great threat to those plants, as K^+^ availability is positively related to plant resistance to damage from external stress [11,31,181,234].

Plants can respond to drought by improving their efficiencies of K^+^ and water uptake [244]. Hydraulic lift and its opposite, downward-siphoning [245], can facilitate water recirculation, water use efficiency, and nutrient uptake capacity [246]. A higher water redistribution capacity to increase K^+^ uptake can be a key factor in the survival of plants under drier conditions.

However, owing to a lack of research, it is not possible to assess the importance of water redistribution in enhancing not only water use efficiency but also the capacity for nutrient uptake. Moreover, we know that drought increases K^+^ retention in the wood of some species and K^+^ concentration in the photosynthetic tissues of others [4]. The effects of drought are typically asymmetrical in different plant organs because plants tend to allocate and retain more K^+^ in stems, and a fraction of this K^+^ is transported to leaves during the driest seasons [4]. However, some studies have observed that the accumulation of K^+^ in soil is greater in arid ecosystems than that in mesic ecosystems, due to K^+^ being more quickly leached from litter than N or P, and thus has a much shorter residence time in soil organic matter [246,247].

### 6.3. Human Impacts on the K+ Potassium Cycle

K^+^ concentrations in soil solution and runoff are more sensitive to changes in land use and fertilization than N or P concentrations; K^+^ has great capacity to spread through the environment [248,249]. The distinct capacity of different plant species to use and remobilize K^+^ can be crucial in their competitive relationships [250,251,252,253]. The increasing use of K^+^ fertilizer in some regions in recent decades has increased the atmospheric deposition of K^+^ in areas adjacent to croplands through the aerial transport of fine soil particles enriched in K^+^ [254]. Although the deposition of K^+^ from energy production and industrial plants has decreased in some European countries [255], that of K^+^ from agricultural activities is very high in some developing countries, such as Brazil, reaching 55% of total K^+^ deposition [256]. K^+^ deposition linked to industrial and mainly agricultural activities can be higher than deposition from natural processes [257,258], such as sea spray [259,260]. The inputs of K^+^ from atmospheric deposition have a determinant role in K^+^ balances of forests, counteracting the loss of K^+^ from leaching, which is favored by N deposition, due indirectly to soil pH decrease and soil degradation and directly to the competition with ammonium for the occupation of the exchangeable complex [261,262]. These inputs of K^+^ contribute to K^+^ availability in tropical forest ecosystems [263,264]. The aforementioned findings warrant further research into the impacts of increased use of K fertilizers on the rates of K^+^ atmospheric deposition and on ecosystem function.

## 7. K and Food Security and Human Health

### 7.1. K Fertilizers and Food Security

The limiting role of K^+^ in agriculture also warrants consideration. First, many studies [45] have reported substantial improvements in crops after an adequate application of K fertilizers via both a direct positive effect on crop production and indirect effects of increasing water-, N-, and P-use efficiencies, preventing disease, and managing stress responses to drought [265,266,267], frost [268], and salinity [269,270,271]. Second, the global demand for K fertilizer has increased in recent years [272]; only some countries, such as China and the USA, control most of the K^+^ production, while several developing countries, such as India and other Asian countries with limited K^+^ fertilization, are expected to drastically increase their consumption in the near future [272]. However, not all countries have adequate resources because only a few minerals, such as mainly evaporite rocks, have the capacity to release K^+^ for use in K^+^ fertilizers, despite the very high abundance of potassium in the lithosphere [272]. This information is even more relevant in the context of global change, in which most future projections report that several large regions of the world, such as the Sahel and areas with Mediterranean climates, will become drier in the next decades [238].

Moreover, the rising demand for food is frequently associated with increasing aridity in several areas of the world, such as the Mediterranean Basin (mainly the Maghreb) and the Sahel, making K^+^ fertilization more critical, given the important role of K^+^ in water-use efficiency [273,274]. A recent study concluded that K^+^ is currently the most important limiting nutrient in Mediterranean croplands [275].

Data from the last decades suggest that some important cropland areas, such as China, Egypt, Bulgaria, and parts of southeastern Asia, also have negative balances of K^+^; this is mainly due to socioeconomic factors that impoverish crop soils in K^+^, with a consequent decrease in crop productivity [275,276,277]. Agricultural practices that decrease soil pH favor K^+^ leaching and thus decrease availability for plants [278]. For example, despite N fertilization, sometimes enhancing soil K^+^ availability in the short term by decreasing soil pH [279]; in the absence of potassium fertilization, N fertilization leads to a decrease in soil K^+^ availability due to leaching [279,280,281]. The demand for and the use of fertilizers are expected to increase in the coming decades [282]. Moreover, depletion of plant-available K^+^ in soils after several years of limited potassium fertilization is frequently observed in India and other developing countries; this leads to a variety of negative impacts, including preventing optimum utilization of applied nitrogen and phosphorus fertilizers and decreasing farmers’ incomes [283].

Mineable reserves of potassium have been considered sufficient to meet projected demand for centuries [282]. However, recent data on fertilizer use at the global scale have questioned the sufficiency of potassium-mineable reserves. Global potassium fertilizer applied to cereal crops increased threefold between 1961 and 2015, and the mineable potassium recently estimated to last 100 years could last less with the current tendency to increase annual potassium fertilizer use, mainly in cereal crops [284]. In this regard, despite being less described than that of the case of N and P, the atmospheric K deposition from human activities represents higher amounts than that of natural sources in some areas [4]. Despite the values of atmospheric K deposition being considerably high in some regions, its variability along space is very high as well [285,286]. K atmospheric deposition mainly originates from agricultural and industrial activities such as fertilization, the cement and steel industry, and coal combustion [287,288,289], but the potential effects of K deposition on terrestrial ecosystems remain to be widely studied. The potential problem of K^+^ leaching due to the massive use of K^+^ fertilization of croplands can also be substantial [290,291]. However, equally for K atmospheric deposition, the impacts of this K^+^ leaching toward other noncrop terrestrial ecosystems and terrestrial body waters has been scarcely studied.

### 7.2. K Fertilization, Food Potassium Content, and Human Health

K^+^ intake is linked to human health [292]. An adequate supply of K^+^ to crops is key to the production of high-quality food [293,294,295]. Therefore, an adequate supply of potassium fertilizer is needed not only to produce food with sufficient K^+^ concentration for an equilibrated human diet [294,295,296,297] but also to improve food quality, as K^+^ availability is associated with a rise in concentrations of sugars [294], antioxidants such as phenolics [298], proteins [299], and oils [299] in food crops. However, the increase in crop overfertilization with potassium poses risks to human health. The daily per capita intake of K^+^ recommended by the World Health Organization (WHO) is 3.51 g; whereas, for example, the estimated K^+^ intake in the Ghanaian population was 9.1 during 2010–2011 (2.6-fold larger than the recommended value) [300]. Excessive intake of K^+^ by humans leads to hypertension and cardiovascular diseases [300]. Deficiency of potassium in humans is rare due to its presence in a great array of foods [301]. However, in certain circumstances, such as an excess consumption of diuretics or during high levels of diarrhea or vomiting, a person can have hypokalaemia that, if detected, can be easily solved by Mg and K supplementation and/or adequate diet [302]. Moreover, a decrease in potassium concentration in basic foods has been detected in some countries, such as the USA, South Korea, and Mexico, where people living in food-insecure households owing to low income and poor diets and low consumption of fruits and vegetables have health problems associated with potassium deficiency [303,304,305,306]. Adequate levels of K^+^ intake in humans are associated with a decreased risk of several important pathologies, such as high blood pressure [307,308] and cardiovascular [307,308] and renal diseases [307]. This constitutes more evidence of the importance of an adequate potassium fertilization management for human health.

When crops grow in soils with poor K^+^ availability, they have reduced photosynthetic rates per unit leaf area; this leads to an overall reduction in the amount of photosynthetic assimilates available for growth and reduced assimilate transport out of the leaves to the developing fruit [293]. However, although less reported, high levels of potassium fertilizer application can also have detrimental effects on crop yields and quality [309]. For example, some cropland areas in China have been overfertilized with potassium during the last 30 years without obtaining higher crop yields and qualities, instead exhibiting a drop in potassium-use efficiency [253,310]. Moreover, an excess of K fertilization can produce a detrimental effect on plant performance and growth by inhibiting the uptake of other nutrients such as N, magnesium, or iron as a result of the competition for the root absorption mechanisms [311,312].

One method for improving potassium-use efficiency in crops and improving K reservation may be to select crop varieties able to maintain higher K^+^ concentrations in cells and to confer more resistance to drought to the extent possible [313]. For instance, the barley K^+^-stress-tolerant genotype “LK tolerant” can adapt to low K^+^ availability through its capacity to shift its metabolism toward the lower presence of negatively charged amino acids, such as Asp and Glu, and higher concentrations of positively charged amino acids (Lys and Gln), and lower carbohydrate consumption to have more energy to grow in K^+^-limited conditions [314]. Recently, it has been found that the overexpression of several genes related to K^+^ transporters and channels can increase the tolerance to K^+^ deficiency. Among these, we can highlight the gene OsPRX2, which causes stomata closure [315]; the gene HAK, which is directly related to K^+^ uptake under low soil K^+^ availability [220]; or genes related to root elongation favoring higher K^+^ uptake [316]. Thus, there is a great opportunity to genetically improve several crop species to improve potassium-use efficiency and help solve the problem of K^+^ scarcity in soils and the access to potassium fertility in future markets [123,272].

## 8. Conclusions

The physiological–metabolic role of K^+^ at the individual plant level is related to the internal transport of substances and energy and to the capacity to respond to biotic and abiotic stresses. This plant-level role also helps in understanding the key role at the community and ecosystem levels. Furthermore, it is also important to introduce the human dimension, because human activities release enormous amounts of K^+^ to the environment, with several consequences to K^+^ availability in natural ecosystems and crops, including effects on plant function and growth that further influence food security and human health and wellness. These three levels of K^+^ roles, plant, ecosystem and human-related, can be summarized as follows:

1. A wide array of K^+^ functions in plants are central to plant homeostatic control, the internal transport of substances and energy, the mechanisms of response to biotic and abiotic stresses, and plant growth and metabolism control. These functions are performed by membrane K^+^ transporters that act within a cascade of processes involving the control of gene transcription and physical (membrane hyperpolarization–depolarization, osmotic changes, pH control) and chemical (cellular ROS, Na^+^, O_2_, and H_2_O) “sensors,” all of which are also associated with the up- and downregulation of the main plant hormones (auxin, ABA, JA) (Figure 1 and Figure 2). Hence, further studies coordinating K^+^ availability; stressful conditions (e.g., drought); plant interactions with abiotic and biotic stresses and changes in field conditions; and metagenomic, transcriptomic, and metabolomic analyses altogether are warranted to obtain a more complete and general understanding of K^+^ movements and functioning of K^+^ transporters in intra- and intercellular membranes. The advancement of the knowledge of how chronic and pulse changes in the availability and fluxes of K^+^ can interact with the main drivers of global change, and also how they can affect ecosystem structure and function along natural gradients, warrant in-depth research.

2. These key roles of K^+^ are consistent with the comparable K^+^ concentrations to those of other fundamental bioelements, such as N and P in the most active plant tissues, although K^+^ is not a structural component of biomolecules. Recent ecological studies demonstrate the key role that K^+^ plays in basic plant and ecological traits in global terrestrial ecosystems (e.g., growth, competition, limitation, photosynthesis, water-balance control, and defense) at the same level of the corresponding roles of N and P. Thus, we highlight the necessity to include K^+^ in all biogeochemical and stoichiometric studies with an ecological point of view, in the studies of ecosystem responses to the drivers of global change, and even in current climatic and Earth-system models.

3. Potassium fertilization with industrial fertilizers from mineable potassium reserves has continuously increased since the Industrial Revolution. Currently, we are in a scenario with a complex dilemma in which rich countries tend to overfertilize with potassium, implying environmental problems and even potential human health risks, whereas poor countries frequently have problems with access to potassium fertilizers, limiting their crop production. This dichotomy occurs under a scenario of increasing limitation of mineable potassium sources, together with a rise in potential demand by crop intensification and aridity increase under climate change. Certainly, the increasing extension of arid and semiarid areas is a problem to consider for food security, since, as mentioned above, water uptake, transport, and use efficiency are among the most relevant functions of K^+^ in plants, and more scientific, technical, and economic efforts are urgently needed to address future access to K^+^ in regions most threatened by increasing aridity and food security (e.g., the Sahel, areas with Mediterranean climates, and parts of Asia and South America). A correct supply of K^+^ fertilization can help to counteract increasing aridity related to food production. On the contrary, an increasing difficulty in accessing to fertilizers together with stronger and more frequent torrential rainfalls and higher aridity can erode soil and increase the leaching of K^+^, thus driving a soil K^+^ impoverishment that could worsen the effects of aridity on food security in several parts of the world, in most cases coinciding with current poor regions.

## Figures and Tables

**Figure 1 plants-10-00419-f001:**
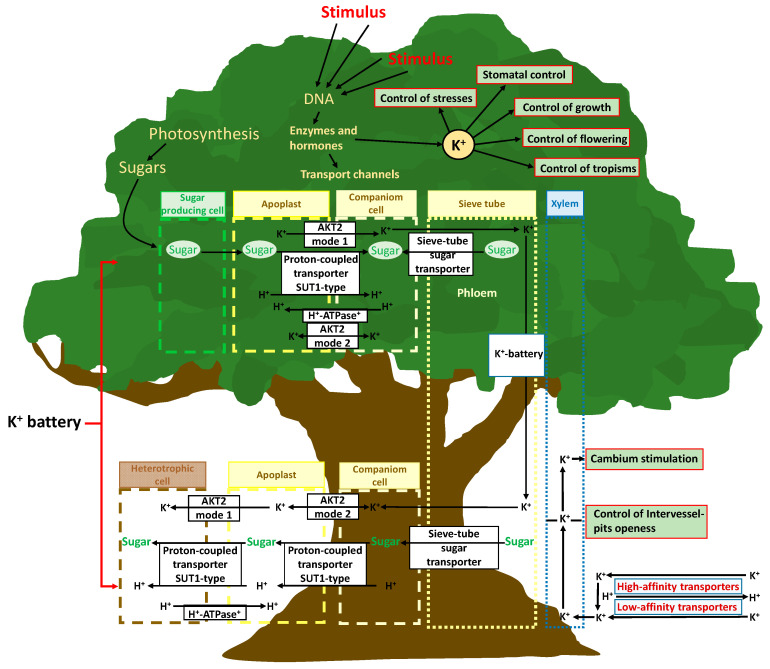
Schematic of the role of K^+^ in plant antistress responses. The transcriptional and post-transcriptional control of K^+^ passive and active membrane transporters is a central topic of current research, owing to their important role in plant development, functionality, and control of responses to environmental stresses that they activate. However, several other parallel stress mechanisms allow K^+^ replenishment and, thus, enhance stress tolerance. Further, external or internal stimuli (e.g., drought, changes in cellular osmotic pressure or pH, changes in metabolite concentrations, or excess light or oxidation) mediate responses by activating genes encoding hormones and/or proteins that can catalyze the transcription of genes that produce high-affinity K^+^ channels. An adequate supply of potassium allows a reduction of carbohydrates in leaf cells by counterbalance mechanisms between K^+^ and organic osmolites in cells. Carbohydrates, mostly in terms of hexose content, are decreased in leaves due to a sufficient K^+^ supply and transported to another plant organ, owing to better phloem activity. AKT2 transporters can operate in mode 1 as an inward-rectifying channel, or as mode 2 as a nonrectifying channel.

**Figure 2 plants-10-00419-f002:**
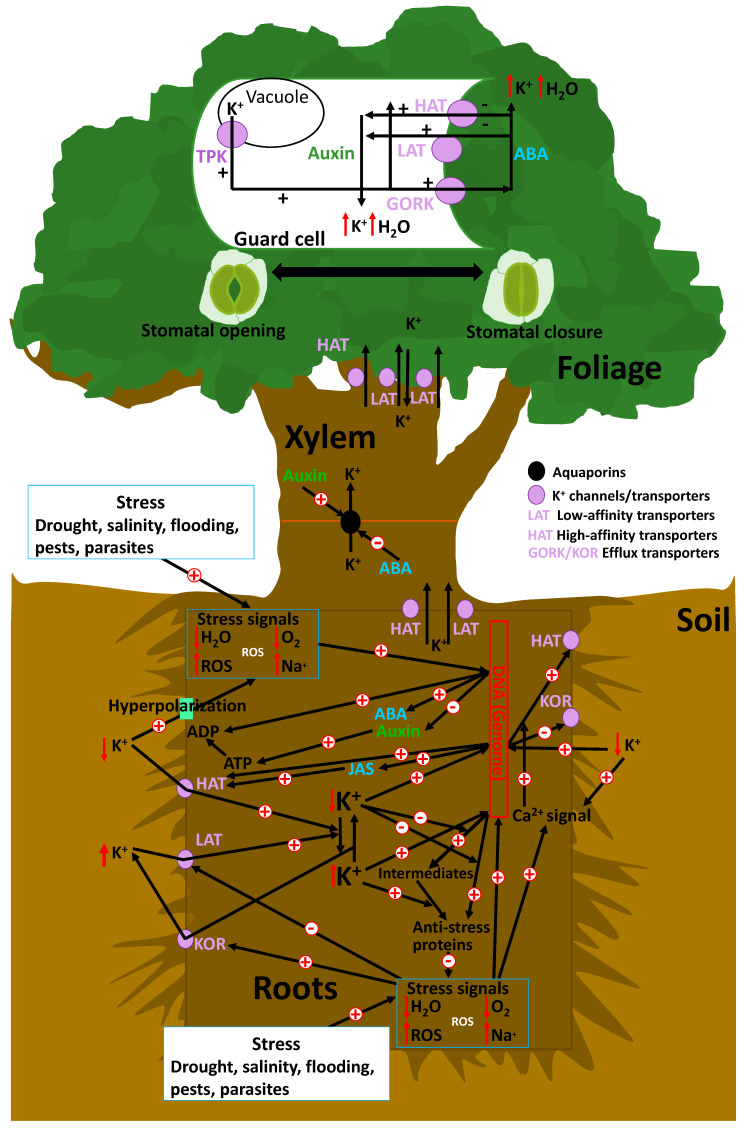
Schematic of the K^+^ uptake and transport in plants, with special attention paid to the potassium battery. The main functions are depicted in green rectangles.

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
