# Peer review of "Potassium Control of Plant Functions: Ecological and Agricultural Implications"

_plants, 2021, doi:10.3390/plants10020419_

Round 1

Reviewer 1 Report

In the present version, the manuscript is improved well. Only one comment;

In Figure 2, a black oval is explained as “Aquaporins”, and is displayed under “Xylem”. This black oval in the tree shows K+ transport and is regulated positively from shoot and negatively from roots by ABA. This looks not an aquaporin.

Relationship between K+ and aquaporins seems to be described adequately in the main text, but these description looks inconsistent with Figure 2.  

Author Response

Response: Done as requested.

Responses to reviewer 1 comments

1.Open Review

English language and style

( ) Extensive editing of English language and style required
( ) Moderate English changes required
( ) English language and style are fine/minor spell check required
(x) I don't feel qualified to judge about the English language and style

Yes

Can be improved

Must be improved

Not applicable

Does the introduction provide sufficient background and include all relevant references?

(x)

( )

( )

( )

Is the research design appropriate?

( )

( )

( )

(x)

Are the methods adequately described?

( )

( )

( )

(x)

Are the results clearly presented?

( )

( )

( )

(x)

Are the conclusions supported by the results?

( )

( )

( )

(x)

Comments and Suggestions for Authors In the present version, the manuscript is improved well.

Response: Thanks for the favourable general evaluation of our study.

2.Only one comment; In Figure 2, a black oval is explained as “Aquaporins”, and is displayed under “Xylem”. This black oval in the tree shows K+ transport and is regulated positively from shoot and negatively from roots by ABA. This looks not an aquaporin. Relationship between K+ and aquaporins seems to be described adequately in the main text, but these description looks inconsistent with Figure 2.  

Response: we have now corrected this mistake. It represents the SKOR channels that allow the transport of K+ from root to xylem.

Reviewer 2 Report

Comments and Suggestions for Authors

In the manuscript entitled “ Potassium Control of Plant Function: Ecological and Agricultural Implications”, the authors provided an a comprehensive overview of Potassium and its complex roles in terrestrial ecosystem functions and in the food security within the current context of climate change.

The half page Introduction is not very attractive. Please use some key words to catch the attention of readers with recent bibliography.

 It should also has to include few lines about……….what is the background of this study and why you choose Potassium (K) to study and why it is important at this stage for individual and at whole ecosystem level.

Authors should provide a clear hypothesis between what is already known and what was already published by them about Potassium in their previous article in 2015……  

Sardans, J. and Peñuelas, J., 2015. Potassium: a neglected nutrient in global change. Global Ecology and Biogeography, 24(3), pp.261-275.

https://doi.org/10.1111/geb.12259

what happens in the last 5 years that was not covered in 2015….gaps and how authors have addressed those gaps in the present article.

In many areas, it looks like a book. Authors should synthesize the literature and add some Tables with summary of important data in different headings.

Lines 56 -59: These lines need to be revised……Meaning is not clear.

Lines 60 -62: These lines need to be revised……Meaning is not clear…. the links of different levels of K+ roles

Lines 621 -625: These lines need to be revised……Meaning is not clear…. PLease Split into two sentences.

Please add sub-headings in each main heading to increase visibility and clarity….Otherwise, the present article structure is not very attractive and readers can lost during Reading the lengthy paragraphs in each main heading.

Lines 653 -656……….and Lines 661 – 664….the same information is repeated. Please delete the aformentioned lines from the Ms.

Lines 671……Litter decomposition of which plant/crop ¿??

Lines 672 -673….This sentence is not complete ¿??

Lines 720….Revise this sentence.

Lines 731-733….Revise this sentence….meanings are not clear.

Lines 744-746….Revise this sentence….meanings are not clear.

Lines 815 -816….Revise this sentence….meanings are not clear.

Lines 853….Replace capital I with small alphabet.

The discussion section should be more relevant to the presented data and not general information should be presented. The Discussion needs to be more focused on the possible reasons for the findings of this study.

Conclusion. The Conclusion needs to be rewritten. It is a mere repetition of the findings of the paper. Authors should mention a clear picture of key processes that might be affected due to Potassium at the molecular, ecological and global scale.

 General comments: the paper should be checked for typos, there are many in the pdf version. English needs to be improved. At some point, it is hard to understand.

Author Response

Responses to reviewer 2 comments

1.Open Review

English language and style

( ) Extensive editing of English language and style required
(x) Moderate English changes required
( ) English language and style are fine/minor spell check required
( ) I don't feel qualified to judge about the English language and style

Yes

Can be improved

Must be improved

Not applicable

Does the introduction provide sufficient background and include all relevant references?

( )

( )

(x)

( )

Is the research design appropriate?

( )

( )

(x)

( )

Are the methods adequately described?

( )

( )

(x)

( )

Are the results clearly presented?

( )

( )

(x)

( )

Are the conclusions supported by the results?

( )

(x)

( )

( )

Comments and Suggestions for Authors

Comments and Suggestions for Authors. In the manuscript entitled “ Potassium Control of Plant Function: Ecological and Agricultural Implications”, the authors provided an a comprehensive overview of Potassium and its complex roles in terrestrial ecosystem functions and in the food security within the current context of climate change.

Response: Thanks for the positive interpretation and evaluation of our manuscript.

2.The half page Introduction is not very attractive. Please use some key words to catch the attention of readers with recent bibliography.

Response: We have followed your insightful suggestion. The revised text now reads:

1. Introduction

In the last years the role of potassium in terrestrial ecosystems has become more studied, thus observing that its role in determining plant growth, species composition and ecosystem function can be of similar importance than the role  of the much more studied elements Nitrogen (N) and phosphorus (P) [1-5]. Differently than N and P, that are mostly present forming part of bio-molecules, K is present in living organisms mostly as free cation (K+). K+ together with calcium (Ca2+) are the two most abundant inorganic chemicals in plant water cellular media and K+ is the second most abundant nutrient in leaf biomass after N, which  highlights its great involvement and unavoidable contribution to plant functioning. At the plant community level, K+ also limits community growth [4,5]. Recent reports have observed great direct impact of potassium in plant photosynthetic capacity [6,7] and growth [8,9] in complex plant functional mechanisms in responses against different stresses [10-13] and plant homeostasis [14] and metabolic control [15]. Since our past revision on the role of K+ in terrestrial ecosystem responses to global change drivers [4] an exponential number of reports allows to establish a better link both between potassium role to the responses to global change drivers [16-19] but also with terrestrial ecosystems function and structural variables such as growth and nutrient cycling [8,9,20,21]. We here synthesize these important roles from the metabolic and physiological functions in individual plants to the complex roles in terrestrial ecosystem functions and food security while considering ongoing global changes. We thus provide a bridge between the studies of K+ in plants and the studies of K+ in ecosystems, to ultimately claim that its introduction in terrestrial ecological studies should be at least at a level similar to those of N and P.”

3.It should also has to include few lines about……….what is the background of this study and why you choose Potassium (K) to study and why it is important at this stage for individual and at whole ecosystem level.

Response: Done, see response to comment 2.

4.Authors should provide a clear hypothesis between what is already known and what was already published by them about Potassium in their previous article in 2015……  Sardans, J. and Peñuelas, J., 2015. Potassium: a neglected nutrient in global change. Global Ecology and Biogeography, 24(3), pp.261-275.https://doi.org/10.1111/geb.1225. what happens in the last 5 years that was not covered in 2015….gaps and how authors have addressed those gaps in the present article.

Response: Done, see response to comment 2

5.In many areas, it looks like a book. Authors should synthesize the literature and add some Tables with summary of important data in different headings.

Response: To make a brief synthesis and a more accurate organization of the whole manuscript without enlarging it, we have added sub-headings (as also advice by the referee in his/her comment 9) throughout the manuscript and provide an index summary after the abstract. This index summary now reads:

Index 

1.Introduction………………………………………………………………………………...   3

  1. Stable cation in solution necessary for plant functional

homeostasis and production control……………………………………………………….   4

            2.1. The biogeochemical properties of K+………………………………………….   4

            2.2. The multiple functions of K+ in plants………………………………………...   5

  1. Plant K uptake mechanisms……………………………………………………………… 6

3.1. The role of transporters and channels……………………………………………..   6                                                                  

3.2. Plant morphology and ecological aspects in plant K+ uptake……………………..    8

  1. Internal transport of K+, water, nutrients, and biomolecules…………………………. 13

4.1. The control of internal water transport…………………………………………...  13

            4.2. The K+ role in the complex mechanisms of internal biomolecules,

nutrients and energy in plants: “the potassium battery”……………………………..  14

  1. Role of K in plant responses to external environmental conditions…………………. 15

            5.1. Abiotic stress……………………………………………………………………...  15

            5.2. Biotic stress……………………………………………………………………….  22

  1. Role of K in terrestrial ecosystems……………………………………………………… 23

6.1. Ecological aspects of K+ cycle in plant-soil system……………………………….  243                                                                                                                                              

            6.2. K+ role in plant-soil system shifts under distinct abiotic

            and biotic environmental circumstances………………………………………………  24

6.3. Human impacts of K+ Potassium cycle…………………………………………..  28

  1. K and food security and human health………………………………………………… 29

7.1. K-fertilizers and food security……………………………………………………   29

            7.2. K-fertilization, food potassium content and human health………………………   31

  1. Conclusions……………………………………………………………………………… 32

References…………………………………………………………………………………...   35

6.Lines 56 -59: These lines need to be revised……Meaning is not clear.

Response: These lines aimed to highlight that despite K+ is not directly involved in the structure of biological molecules (from DNA or lignin to sugars), its presence in cell media is comparable to other bio-elements/nutrients very important in the structure of the most important groups of bio-molecules such as proteins or genetic material, which, as said, highlights its potential contribution to plant functioning. We have rewritten this text to clearly transmit this message. The revised text now reads:

 “Differently than N and P, that are mostly present forming part of bio-molecules, K is present in living organisms mostly as free cation (K+). K+ together with calcium (Ca2+) are the two most abundant inorganic chemicals in plant water cellular media and K+ is the second most abundant nutrient in leaf biomass after N, which highlights its great involvement and unavoidable contribution to plant functioning”

7.Lines 60 -62: These lines need to be revised……Meaning is not clear…. the links of different levels of K+ roles

Response: Now clarified. Please see revised text in response to comment 2.

8.Lines 621 -625: These lines need to be revised……Meaning is not clear…. PLease Split into two sentences.

Response: Rewritten from:

In general, some Inceptisols and Entisols are particularly rich in clays and not excessively leached, Alfisols rich in clay content and also not excessively leached and Mollisols for its high humus concentration with very large nutrient retention capacity are the soils most likely to have the highest K+ availability [241].

To:

In most cases Inceptisols and Entisols are particularly rich in clays and not excessively leached. Alfisols rich in clay content and also not excessively leached, and thus can have good K+ availability, whereas Mollisols for its high humus concentration with very large nutrient retention capacity are the soils most likely to have the highest K+ availability [241].”

9.Please add sub-headings in each main heading to increase visibility and clarity….Otherwise, the present article structure is not very attractive and readers can lost during Reading the lengthy paragraphs in each main heading.

.

10.Lines 653 -656……….and Lines 661 – 664….the same information is repeated. Please delete the aformentioned lines from the Ms.

Response: Done. We have now avoided this unnecessary repletion. The revised text now reads:

The role of K+ in several ecological and ecophysiological processes can be also influenced by N eutrophication, increasing atmospheric CO2, or changes in land cover, but K+ itself can play a significant role in the response/adaptation of terrestrial ecosystems to these global change drivers. This possible key role remains to be studied at local, regional, and global scales [4]. The sophisticated mechanisms in which K+ is involved are pivotal for water and nutrient use efficiency, stress avoidance, and general plant homeostasis control, altogether implying several ecological consequences for the ecosystem, such as species replacement or altered water and nutrient cycling. For example, in the drought-tolerant tree species Olea europea when soil temperature reaches 37 ºC, its root growth is inhibited and also K+ uptake with a drop of stem K+ concentration, water content and use efficiency [248].”

11.Lines 671……Litter decomposition of which plant/crop ¿??

Response: Now clarified. The revised sentence now reads:

As observed in several forests, such as temperate and tropical forests, K+ is rapidly released during litter decomposition [249-251].”

12.Lines 672 -673….This sentence is not complete ¿??

Response: Now completed:

However, more than 70% of the initial K+ content can be released during the first week of the decomposition process [252].”

13.Lines 720….Revise this sentence.

Response: Revised to now read:

Most macroecological studies identifying patterns of foliar nutrient concentrations as a function of global environmental traits have focused on N and P and their ratios [281,282]. Far fewer studies have focused on K [4].”

14.Lines 731-733….Revise this sentence….meanings are not clear.

Response: Clarified. It now reads:

Climate models project an increase in the extent of drought over large areas of the world such as the Mediterranean Basin and the Sahel [289].”

15.Lines 744-746….Revise this sentence….meanings are not clear.

Response: Clarified. It now reads:

Thus, current data suggest that K+ uptake is strongly determined by soil water availability [292,293] and also the other way around, that K+ is crucial to the strategies of water uptake and economy by increasing water use efficiency and limiting water loss.”

16.Lines 815 -816….Revise this sentence….meanings are not clear.

Response: Clarified. It now reads:

The demand and the use of fertilizers are expected to increase in the coming decades [338].”

17.Lines 853….Replace capital I with small alphabet.

Response: Replaced.

18.The discussion section should be more relevant to the presented data and not general information should be presented. The Discussion needs to be more focused on the possible reasons for the findings of this study.

Response: The entire manuscript, except for the introduction section that has now been rewritten in response to reviewer’s comment, is a discussion based on the main most recent reports about the role of K+ in terrestrial plants and ecosystems, and finally in human food security and health. We have now, organized with more detail all the text (see the responses to comments 5 and 9).

19.Conclusion. The Conclusion needs to be rewritten. It is a mere repetition of the findings of the paper. Authors should mention a clear picture of key processes that might be affected due to Potassium at the molecular, ecological and global scale.

Response: Done. The revised text now reads:

7. Conclusions

The physiological-metabolic role at K+ at the individual plant level is related to the internal transport of substances and energy and to the capacity to respond to biotic and abiotic stresses. This plant level role helps to understand the key role also at the community and ecosystem level. Furthermore, it is also important to introduce human dimension because human activities release enormous amounts of K+ to environment with several consequences on K+ availability on natural ecosystems and crops including effects on plant function and growth that further influence on food security and human health and wellness. These three levels of K+ roles, plant, ecosystem and human-related can be summarized as follows:

  1. A wide array of K+ functions in plants are central to plant homeostatic control, the internal transport of substances and energy, the mechanisms of response to biotic and abiotic stresses, and plant growth and metabolism control. These functions are performed by membrane K+ transporters that act within a cascade of processes involving the control of gene transcription and physical (membrane hyperpolarization–depolarization, osmotic changes, pH control) and chemical (cellular ROS, Na+, O2, and H2O) “sensors,” all of which are also associated with up- and down-regulation of the main plant hormones (auxin, ABA, JA) (Figures 1 and 2). Hence, further studies coordinating K+ availability, stressful conditions (e.g., drought), plant interactions with abiotic and biotic stresses and changes in field conditions, and metagenomic, transcriptomic, and metabolomic analyses altogether are warranted to obtain a more complete and general understanding of K+ movements and functioning of K+ transporters in intra- and intercellular membranes. The advance of the knowledge of how chronic and pulse changes in the availability and fluxes of K+ can interact with the main drivers of global change and also how they can affect ecosystem structure and function along natural gradient warrants in-depth research.
  2. These key roles of K+ are consistent with the comparable K+ concentrations to those of other fundamental bio-elements, such as N and P in the most active plant tissues, although K+ is not a structural component of biomolecules. Recent ecological studies demonstrate the key role that K+ plays in basic plant and ecological traits in global terrestrial ecosystems (e.g., growth, competition, limitation, photosynthesis, water-balance control, and defence) at the same level of the corresponding roles of N and P. Thus, we highlight the necessity to include K+ in all biogeochemical and stoichiometric studies with an ecological point-of-view and in the studies of ecosystem responses to the drivers of global change and even in current climatic and Earth-system models.
  3. Potassium fertilization with industrial fertilizers coming from mineable potassium reserves has continuously raised since the industrial revolution. Currently, we are in a scenario where rich countries tend to over-fertilizer with potassium implying environmental problems and even potential human health risks, whereas poor countries frequently have problems to access to potassium fertilizers limiting its crop production. This dichotomy occurs under a scenario of increasing limitation of potassium mineable sources together with a rise of potential demand by crop intensification and aridity increase under climate change. Certainly, the increasing extension of arid and semi-arid areas is a problem to take into account for food security. since, as above commented, water uptake, transport and use efficiency are among the most relevant functions of K+ in plants, more scientific, technical, and economic efforts are urgently needed to address future access to K+ in regions most threatened by increasing aridity and food security (e.g., the Sahel, areas with Mediterranean climates, and parts of Asia and South America). A correct supply of K+ fertilization can help to counteract increasing aridity regarding to food production. On the contrary, an increasing difficulty of accessing to fertilizers together with stronger and more frequent torrential rainfalls coming together with more aridity can erode soil and increase the leaching of K+, thus driving to a soil K+ impoverishing that can make even worse the aridity effects on food security in several parts of the world, in most cases coinciding with current poor regions.”

20.General comments: the paper should be checked for typos, there are many in the pdf version. English needs to be improved. At some point, it is hard to understand.

Response: We have checked all the typos and the English writing.

We very much thank the consideration and very helpful comments of the editor and the three referees.

Best wishes

Prof. Josep Peñuelas and Dr. Jordi Sardans

Global Ecology unit CREAF-CSIC-UAB

CREAF, Edifici C, Facultat de Ciències

Universitat Autónoma de Barcelona

Round 2

Reviewer 2 Report

The authors have significantly revised and improve the article and now can be accepted for publication.

Author Response

Prof. Snow Liu

Associate Editor

Plants

                                                                                     February 18th, 2021, Barcelona

Dear Prof. Snow Liu:

We submit the revision of our manuscript No. ID: plants-1079934 entitled “Potassium Control of Plant Function: Ecological and Agricultural Implications” after incorporating the comments and suggestions of the editor and the three referees. Many thanks for your consideration and help. We hope that now this study can be accepted for publication in Plants.

Response to Academic Editor Notes

1.Although the authors revised the manuscript I have one important comment:
There are plenty of references cited in this paper. It seems about half of the texts are references!  Please reduce. Keep the references related to K and also remove older references. Also, I suggest making the type 'Review'. I am wondering why and how it is a perspectives article
.

Response: Done as requested.

Response to reviewer 2 comments

1.(x) I would not like to sign my review report
( ) I would like to sign my review report

English language and style

( ) Extensive editing of English language and style required
( ) Moderate English changes required
( ) English language and style are fine/minor spell check required
(x) I don't feel qualified to judge about the English language and style

Yes

Can be improved

Must be improved

Not applicable

Does the introduction provide sufficient background and include all relevant references?

(x)

( )

( )

( )

Is the research design appropriate?

(x)

( )

( )

( )

Are the methods adequately described?

(x)

( )

( )

( )

Are the results clearly presented?

( )

( )

( )

(x)

Are the conclusions supported by the results?

( )

( )

( )

(x)

Comments and Suggestions for Authors

Response: Thanks again to the referee by the favorable evaluation of our study.

2.The authors have significantly revised and improve the article and now can be accepted for publication.

Response: Thanks again to the referee by the favorable evaluation of our study.

We very much thank the consideration and very helpful comments of the editor and the three referees.

Best wishes

Prof. Josep Peñuelas and Dr. Jordi Sardans

Global Ecology unit CREAF-CSIC-UAB

CREAF, Edifici C, Facultat de Ciències

Universitat Autónoma de Barcelona
